# Turn-on chemiluminescence probes and dual-amplification of signal for detection of amyloid beta species in vivo

Jing Yang[1,5], Wei Yin[1,5], Richard Van[2], Keyi Yin[1], Peng Wang[1], Chao Zheng[1], Biyue Zhu[1], Kathleen Ran ![ORCID] [1], Can Zhang[3], Mohanraja Kumar[4], Yihan Shao[2] & Chongzhao Ran ![ORCID] [1✉]

Turn-on fluorescence imaging is routinely studied; however, turn-on chemiluminescence has been rarely explored for in vivo imaging. Herein, we report the design and validation of chemiluminescence probe ADLumin-1 as a turn-on probe for amyloid beta (Aβ) species. Two-photon imaging indicates that ADLumin-1 can efficiently cross the blood–brain barrier and provides excellent contrast for Aβ plaques and cerebral amyloid angiopathy. In vivo brain imaging shows that the chemiluminescence signal of ADLumin-1 from 5-month-old transgenic 5xFAD mice is 1.80-fold higher than that from the age-matched wild-type mice. Moreover, we demonstrate that it is feasible to further dually-amplify signal via chemiluminescence resonance energy transfer (DAS-CRET) using two non-conjugated smart probes (ADLumin-1 and CRANAD-3) in solutions, brain homogenates, and in vivo whole brain imaging. Our results show that DAS-CRET can provide a 2.25-fold margin between 5-month-old 5xFAD mice and wild type mice. We believe that our strategy could be extended to other aggregating-prone proteins.

[1] Athinoula A. Martinos Center for Biomedical Imaging, Department of Radiology, Massachusetts General Hospital/Harvard Medical School, Room 2301, Building 149, Charlestown, Boston, MA 02129, USA. [2] Department of Chemistry and Biochemistry, University of Oklahoma, Norman, OK 73019, USA. [3] Genetics and Aging Research Unit, McCance Center for Brain Health, Mass General Institute for Neurodegenerative Disease, Department of Neurology, Massachusetts General Hospital and Harvard Medical School, Charlestown, MA, USA. [4] Department of Chemistry, Massachusetts Institute of Technology, Cambridge, MA, USA. [5] These authors contributed equally: Jing Yang, Wei Yin. ✉email: cran@nmr.mgh.harvard.edu

Optical imaging has been widely applied in preclinical and clinical studies. Among the optical imaging modalities, near infrared fluorescence imaging (NIRF) is one of the most used technologies for preclinical investigations[1–3]. However, NIRF imaging has several intrinsic limitations that are caused by excitation light, which is needed as input light to excite the imaging probe[2–6]. First, for a biological sample, the excitation light not only excites the fluorescent probe, but also other fluorophore-containing molecules in the sample to cause troublesome autofluorescence. Second, due to the Stokes shift, the wavelength of excitation is always much shorter than the emission wavelength, while the tissue penetration is reversely correlated with the wavelength, therefore the excitation at a relatively shorter wavelength could be problematic in penetrating the biological sample. Third, a fluorescent probe with small Stokes shift suffers from the interference from excitation leakage in real imaging practice due to the imperfection of filters in imaging systems. Fourth, the excitation light generates relatively much stronger emission signals from fluorophores at shallow locations, which contain nonspecifically accumulated imaging probe and auto-fluorescent molecules. These excitation-related limitations of NIRF imaging are significantly contributed to low signal to noise ratio (SNR) and poor tissue penetration. Fortunately, the problems of fluorescence imaging can be partially resolved by chemi or bioluminescence imaging, which do not require external excitation light. Therefore, compared to NIRF imaging, chemi or bioluminescence imaging can provide much better tissue penetration capacity at the same emission wavelength. A recent study demonstrated that 4-cm tissue penetration could be achieved with chemiluminescence imaging at an 800 nm emission, while NIRF imaging could not provide this capacity at the same emission[7].

However, compared to fluorescence imaging, chemiluminescence imaging is being applied far less in biological studies, primarily due to its low sensitivity and irreversibility of the probes. Another drawback of chemiluminescence is the need of a chemical reaction to produce the emission light[8–15]. In most cases the reaction is dependent on stimulation from reactive oxygen (nitrogen) species (ROS). Therefore, most chemiluminescence studies are related to imaging ROS. Nonetheless, some chemiluminescence probes have an auto-oxidation feature, which is not reliant on ROS[16,17]. In this report, a chemiluminescence probe is designed, and we term it as ADLumin-1. We demonstrate that the auto-oxidation of ADLumin-1 can be utilized to dramatically turn on its chemiluminescence in the presence of Aβ species.

Although chemiluminescence has been intensively explored in many diseases such as cancers and diabetes, its applications in brain disorders including Alzheimer's disease (AD) is scarce. Aβ plaques and neurofibrillary Tau tangles are the most pronounced and characteristic hallmarks of Alzheimer's disease (AD)[18], and several imaging methods been widely applied in preclinical and clinical studies[19–26]. Optical imaging, including NIRF imaging and two-photon imaging, are the most used methods for preclinical investigations of fundamental pathological questions and for therapy monitoring in drug development. In the past years, our group has developed a series of smart NIRF probes, CRANAD-X for Aβs[27–31], and numerous other smart (turn-on) fluorescence probes have also been reported[20,26,32–45]. However, turn-on (smart) chemiluminescence probes are rare[46] and none has been applied for studying Aβs. In this work, we report the first smart chemiluminescence probe, ADLumin-1, for Aβ species. We show that an amplification beyond 200-fold in vitro can be achieved. Meanwhile, to overcome the limitation of short emission of ADLumin-1, we demonstrate the feasibility to achieve dual-amplification of signal via chemiluminescence resonance energy transfer (termed as DAS-CRET) with two nonconjugated smart probes in solutions, tissues and brain homogenates, and

in vivo whole brain imaging. We believe that our strategy is not only applicable for detecting Aβs, but also for other aggregating-prone proteins. Moreover, our results indicate that the strategies for turning-on fluorescence could be used for amplifying chemiluminescence, therefore we envision that our studies will inspire considerably more research on chemiluminescence. In addition, we believe that this technology can become a very important complimentary tool for preclinical studies and has potential for clinical studies (via ocular imaging) in the future.

## Results

**Design and synthesis of ADLumin-1.** Several scaffolds have been widely used for generating chemiluminescence, including dioxetane[9,47–49], luminol[50], imidazo[1,2-a]pyrazin-3(7H)-one[51], oxalate[11,13], lucigenin[50], 9,10-dimethyl-anthracene[14], and 10-methyl-9-(phenoxycarbonyl) acridinium[52]. All of these scaffolds have short emission wavelengths (<500 nm) and are not ideal for in vivo imaging, particularly for deep locations such as brains. Moreover, some of the scaffolds have intrinsic limitations for brain imaging. For example, lucigenin and acridinium have charges and are very polar, which can lead to poor brain penetration. Oxalate ester is difficult to extend the wavelength, while dioxetane are not always very stable. In this report, we selected imidazo[1,2-a]pyrazin-3(7H)-one (IPO) as the scaffold (Moiety A in Fig. 1a), due to its easy modifiability for wavelength extension and high stability.

Our probe design is expected to meet the following requirements. First, the designed compounds should bind strongly to Aβs. In this regard, we surveyed the structures of fluorescence dyes that are sensitive to Aβs, and found numerous dyes containing moiety B (Fig. 1a), which has the potential to insert into the beta sheet of Aβs. Second, the probe should have turn-on capacity upon Aβ binding. Since moiety B is a hydrophobic and planar fragment, it could interact with the hydrophobic segment of Aβs. Consequently, the interaction could lead to turn-on chemiluminescence. Third, the designed probes should have longer emissions. To achieve this, the connection between moiety A and B should allow electrons to delocalize, which can lead to a smaller energy gap between HOMO and LUMO. Based on the above considerations, we designed ADLumin-1 and -2 (Fig. 1a, b), in which moiety A is responsible for generating chemiluminescence and moiety B is for binding to Aβs, and moieties A and B are connected via double bonds that can make the electrons delocalize across the whole molecule. The synthesis of ADLumin-X (X = 1, 2) is very straightforward and the route is shown in Fig. 1c.

**Spectral characterization and validation of the turn-on feature.** Normally, all chemiluminescent molecules are fluorescent. We recorded the excitation and emission spectra of ADLumin-X, and found that the emission peak of ADLumin-1 was around 590 nm (Supplementary Fig. 1) in DMSO, which is considerably longer than most of the commercially available chemiluminescence probes. We had expected the emission of ADLumin-2 to be longer than that of ADLumin-1, due to its additional double-bond. Indeed, the emission peak of ADLumin-2 was 20 nm longer. However, we found that the quantum yield (QY) of ADLumin-2 was significantly decreased ($\Phi_{ADLumin-2}/\Phi_{ADLumin-1} = 0.67$). Based on this fact, we decided to utilize ADLumin-1 for our further investigation, due to its high QY.

ADLumin-1 is stable in PBS buffer and organic solvents, including acetonitrile, dichloromethane, and methanol. However, interestingly, we found that in the presence of 10% DMSO, ADLumin-1 is luminescent, and the emission peak was around 540 nm (Fig. 2a). We speculated that the luminescence was due to

**Fig. 1 Design and synthesis of ADLumin-1. a** Diagram of designing of chemiluminescent probes that contain moiety A for generating chemiluminescence (green) and moiety B for binding to Aβs; **b** Chemical structures of the designed probes ADLumin-X (X = −1 and −2) and diagram for illustrating turn-on with ADLumin-X in the presence of Aβs; **c**) Synthetic route of ADLumin-Xs. Reagents and conditions: (I) TMP, n-BuLi, THF; (II) 5-bromo-2-aminopyrazine, Pd(PPh₃)₄, NaCO₃ aq, 1,4-dioxane; (III) Methylglyoxal 1,1-dimethyl acetal, EtOH, HCl aq.

the auto-oxidation of ADLumin-1. Although the mechanism of chemiluminescence of IPO derivatives has been studied, the mechanism of auto-oxidation has been rarely explored[51]. In the course of our experiments, we noticed that ADLumin-1 emitted very strong luminescence in pure DMSO (Fig. 2b). However, ADLumin-1 showed no apparent chemiluminescence in other solvents, including methanol, ethyl acetate, dichloromethane. Unexpectedly, if the ADLumin-1 DMSO solution was stirred, even stronger luminescence can be detected (3.4-fold in Fig. 2b). We speculated that the luminescence was related to oxygen levels in DMSO. To validate our hypothesis, we intermittently bubbled $O_2$ in the DMSO solution 5 s for several rounds, and found that the intensities were dramatically increased (27-fold) after first bubbling. However, the increasing-fold decreased with cycles (Fig. 2c). LC-MS indicated that ADLumin-1 was converted into a new compound, termed as ADLumin-3 (Fig. 2d), whose structure was further confirmed by $^1$H-NMR, $^{13}$C-NMR, and HR-MS. Since no ROS were added into the DMSO solution, we cautiously concluded that the auto-oxidation of ADLumin-1 was oxygen-level dependent. It is worthy to note that green fluorescent protein and luciferase/luciferin are also dependent on oxygen for their proper imaging functions[53,54].

We also investigated the responses of ADLumin-1 towards different ROS species, and found that the intensity increasing was considerably low, compared to the increasing in the presence of $O_2$ bubbling (Supplementary Fig. 2A, B), again suggesting that the auto-oxidation is the primary cause of chemiluminescence of ADLumin-1. We have proposed a tentative mechanism for the

auto-oxidation in Fig. 2e, in which $O_2$ is added to the double-bond of imidazole moiety to form an unstable intermediate that further decomposes into ADLumin-3 and releases photons (light). Full elucidation of the mechanism needs more intensive investigation.

To investigate whether ADLumin-1 can bind to Aβs, we first recorded the fluorescence spectra before and after the addition of Aβ40 aggregates. As expected, ADLumin-1 was a smart fluorescence probe, evident by a 100-fold fluorescence intensity increasing upon mixing with Aβs (Fig. 3a). Moreover, we found that there was a significant blue-shift of emission of the probe with Aβs (Supplementary Fig. 2A), suggesting the probe binds to the hydrophobic fragment of Aβs. We also measured the Kd via concentration titration, and Kd = 2.1 uM (Supplementary Fig. 2B).

To investigate whether there is a significant chemiluminescence intensity increase upon incubation with Aβs, we tested ADLumin-1 in PBS buffer (with 5% DMSO). Similar to the fluorescence response, the chemiluminescence was dramatically turned-on, and the increase was about 216-fold at 540 nm (Fig. 3b). We also conducted molecule docking studies for ADLumin-1 with Aβ fibrils. The docking was based on the cryo-EM structure of Aβ42 (PDB:5OQV), and the results suggested that ADLumin-1 preferred to bind to the hydrophobic groove formed by Phe19, Ala21, Val24, Asn27, and Ile31 (Fig. 3c). This hydrophobic binding is consistent with the blue-shift of the fluorescence spectrum of ADLumin-1 in the presence of Aβ aggregates (Supplementary Fig. 2A, C). We have also incubated

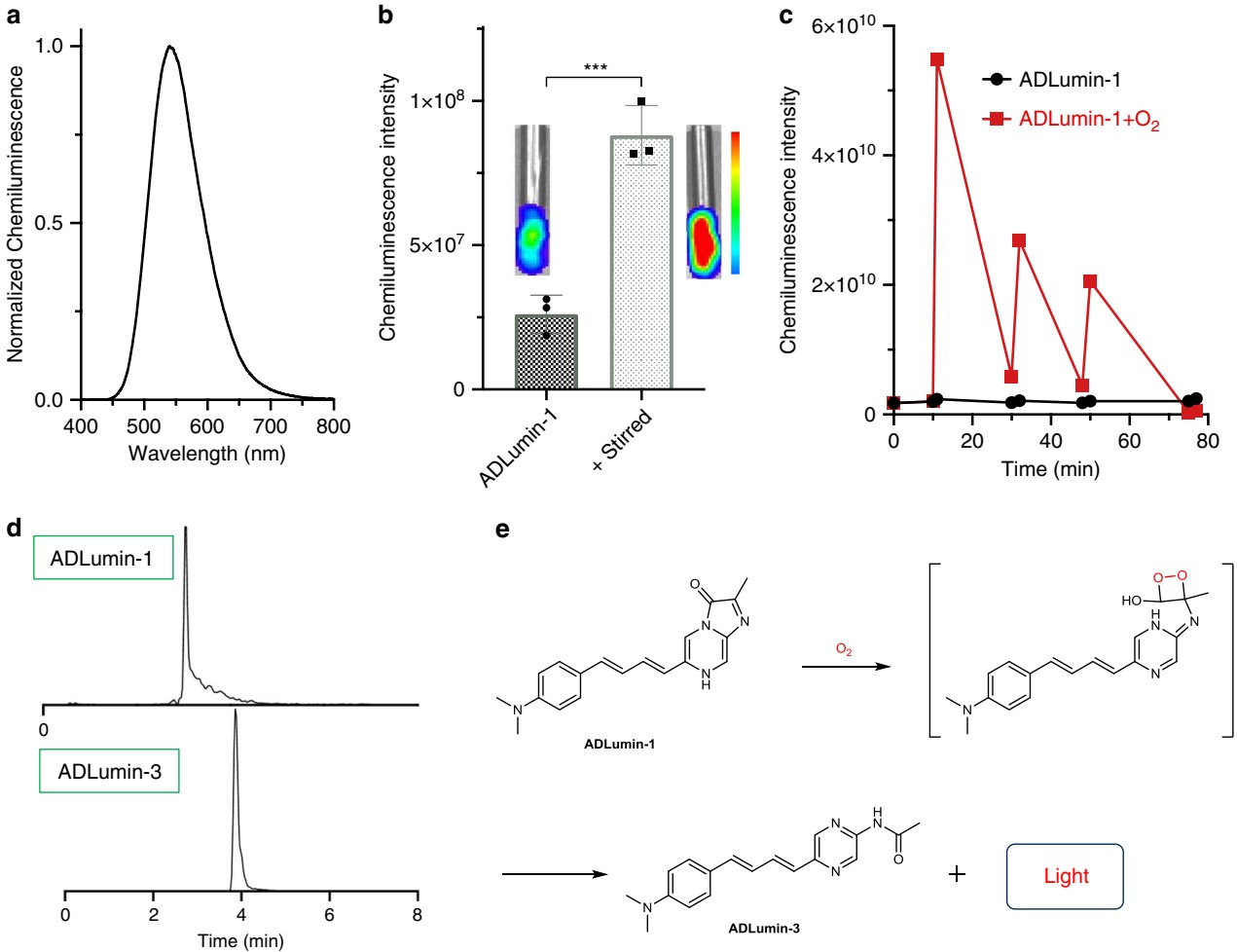

**Fig. 2 In vitro characterization of chemiluminescence properties of ADLumin-1. a** Chemiluminescence spectrum of ADLumin-1 in DMSO; **b** Quantitative analysis of chemiluminescence intensity of ADLumin-1 in DMSO solutions before and after stirring. Representative images were inserted. **c** Quantification of chemiluminescence intensity of ADLumin-1 with (red) or without (black) bubbled oxygen; **d** LC-MS spectra of ADLumin-1 and ADLumin-3 (the oxidation product). Note: ADLumin-1 ($t = 3.05$ min), ADLumin-3 ($t = 3.85$ min); **e** Proposed oxygen-dependent mechanism of ADLumin-1 for chemiluminescence generation. Data are represented as mean ± s.d. with $n = 3$ biologically independent samples. $P = 0.0009$ (**b**). The $P$-values were generated by Graphpad Prism 8 with two-tailed unpaired $t$-test; ***$P$-value < 0.001; Source data underlying **b** and **c** are provided as Source Data file.

ADLumin-1 with Aβ oligomers in PBS solution and found no significant change of chemiluminescence intensity. These results suggested that ADLumin-1 had specific binding with Aβ fibrils.

To investigate whether an IVIS imaging system could be used to detect the increase of chemiluminescence of ADLumin-1 with Aβs, we performed experiments with open filter setting on a 96-well plate. Indeed, the signal was 104-fold higher from the wells with Aβs than that from wells without Aβs (Fig. 3d, e). Though numerous fluorescence probes and smart (turn-on) probes have been reported for Aβs in the past decades, it is worth noting that ADLumin-1, to the best of our knowledge, is the first smart chemiluminescence probe for Aβs.

To explore whether the smart chemiluminescence probe can be used in a real biological environment, we incubated ADLumin-1 with mouse brain homogenate in the presence and absence of Aβ aggregates (12.5 μM). Remarkably we found that ADLumin-1 provided much higher signal from the Aβ group, and the difference was about 11.6-fold (Fig. 3f, g). The intensity increasing was linear to Aβ concentrations in the range of 0–12.5 μM (Supplementary Fig. 3C). Interestingly, MCLA, a commercially available chemiluminescence compound containing IPO moiety, showed significantly reduced signal with Aβs

(Supplementary Fig. 4A, B). The underlying mechanism of the different chemiluminescence responses of MCLA and ADLumin-1 towards Aβ species is still under our investigation.

During the spectral studies, we noticed that the full width at half-maximum (FWHM) of ADLumin-1 was considerably large (94 nm, Fig. 4a). We hypothesized that, for a probe with a relatively short emission peak (540 nm), the larger FWHM could be beneficial for in vivo studies. To validate this hypothesis, we compared ADLumin-1 to widely used firefly luciferin (peak at 570 nm, FWHM = 83 nm), which has been proven to have excellent tissue penetration. In our experiment, we placed a tube that had similar intensity of emitted light from ADLumin-1 or firefly luciferin (with luciferase) (Fig. 4b) under the abdomen of a nude mouse (the depth is about 1.5 cm) (Fig. 4c), and then collected the signals with an IVIS imaging system, which captured the signals from the dorsal side. We found that ADLumin-1 provided comparable signals to firefly luciferin (Fig. 4c, d). There was about 5% of the light that penetrated through the whole body for both ADLumin-1 and luciferin (Fig. 4d). The data supported our hypothesis that the larger FWHM of ADLumin-1, whose emission is shorter than that of firefly luciferin, could be beneficial for in vivo imaging.

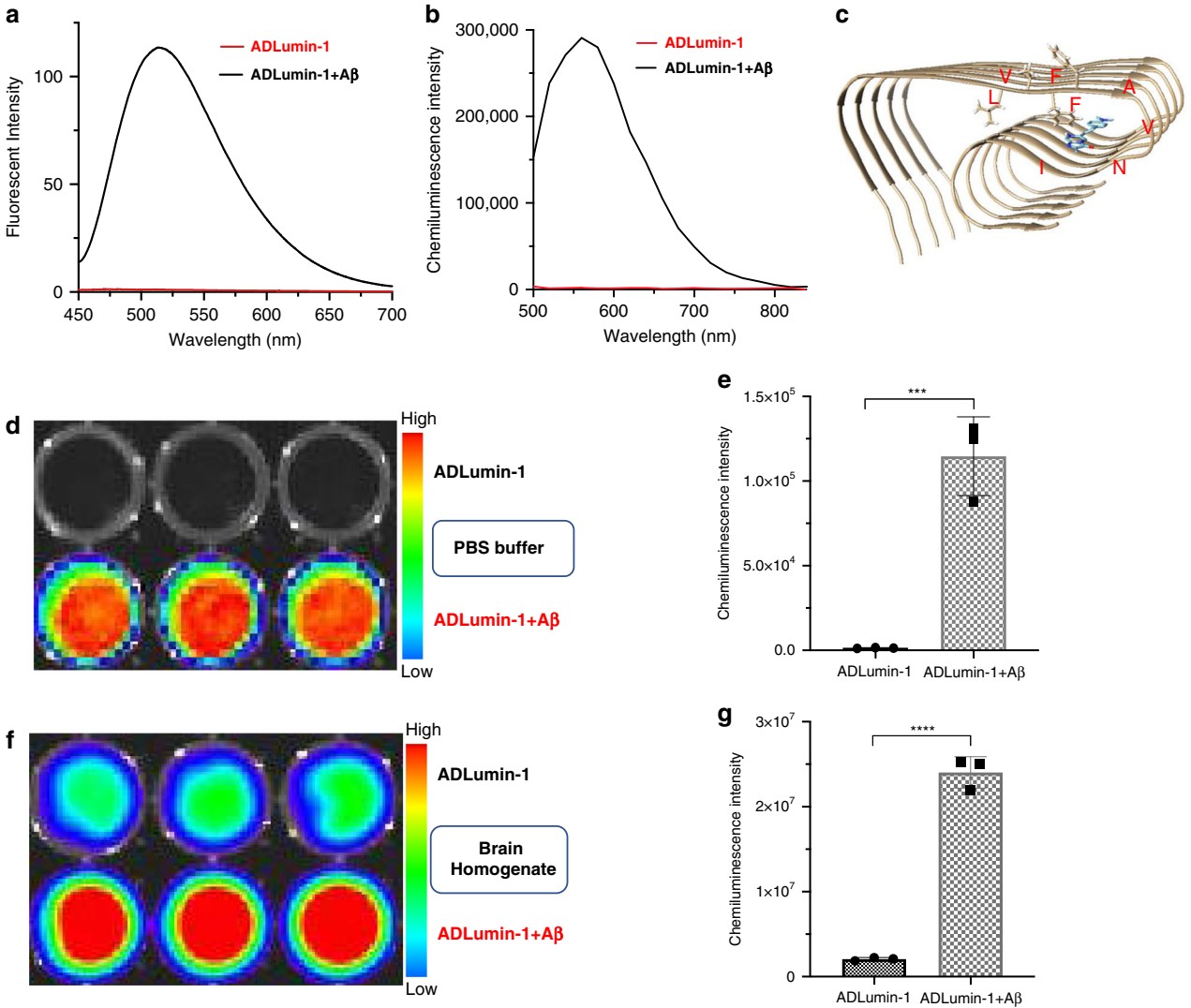

**Fig. 3 In vitro characterization of fluorescence and chemiluminescence properties of ADLumin-1 in the presence of Aβs. a** Fluorescence spectra of ADLumin-1 alone (red) and with Aβ40 aggregates (black); **b** Chemiluminescence spectra of ADLumin-1 alone (red) and with Aβ40 aggregates (black); **c** Molecule docking of ADLumin-1 (blue) with Aβ fibrils (PDB:5OQV). **d** In vitro chemiluminescence images of ADLumin-1 alone and with Aβ40 aggregates in PBS. **e** Quantification of chemiluminescence intensity in **d**. **f** In vitro chemiluminescence images of ADLumin-1 alone and with Aβ40 aggregates in mouse brain homogenate; **g** Quantification of chemiluminescence intensity in **f**. Data are represented as mean ± s.d. with $n = 3$ biologically independent samples. $P = 0.0005$ (**e**), $P < 0.0001$ (**g**). The $P$-values were generated by Graphpad Prism 8 with two-tailed unpaired $t$-test; ***$P$-value < 0.001, ****$P$-value < 0.0001. Source data underlying **e** and **g** are provided as Source Data file.

**In vivo imaging with ADLumin-1**. To verify the binding of ADLumin-1 to Aβs in a real biologically relevant environment, we explored whether ADLumin-1 can stain Aβ plaques in vivo. In this regard, a 5xFAD mouse brain slice was first incubated with ADLumin-1. As expected, ADLumin-1 provided an excellent contrast for the plaques. The signal to noise ratio (SNR) was 2.93 (Fig. 5a, d). To determine whether ADLumin-1 is capable of labeling Aβ plaques in vivo, we conducted two-photon imaging with a 5xFAD mouse, a widely used AD model[55]. We observed that ADLumin-1 could quickly cross the blood–brain barrier (BBB), and provided excellent contrast for cerebral amyloid angiopathy (CAA) on the blood vessels and Aβ plaques (Fig. 5b). The SNR of plaques and CAAs were about 17.0 and 26.0, respectively (Fig. 5e). The time-course imaging results revealed that the intensity in CAA reached the peak within 1 min, while it was 5 min for the plaques. These results suggested that ADLumin-1 could penetrate BBB and stay in the brain

(Supplementary Fig. 5A–C), and it was also an excellent two-photon imaging probe for Aβs in vivo.

To further confirm that ADLumin-1 is indeed able to stain the plaques, we sacrificed the mouse, and sectioned its brain into 20-micron thick slices for histology studies. As expected, the plaques in the brain slice were considerably bright (Fig. 5c, f), confirming that ADLumin-1 could label the plaques in vivo.

To investigate whether ADLumin-1 can be used to detect Aβs in vivo with a noninvasive imaging setup, we used an IVIS imaging system to conduct image acquisition. 4-month old 5xFAD mice ($n = 4$) and age-matched wild-type (WT) were used, and a solution of ADLumin-1 (4 mg/kg) was injected intravenously. Images were captured at 15-, and 30-min after the injection with open filter setting. Although the emission peak of ADLumin-1 is relatively shorter in comparison of other NIRF dyes for brain imaging, we observed considerable strong signals from the brain area, and the differences between 5xFAD and WT

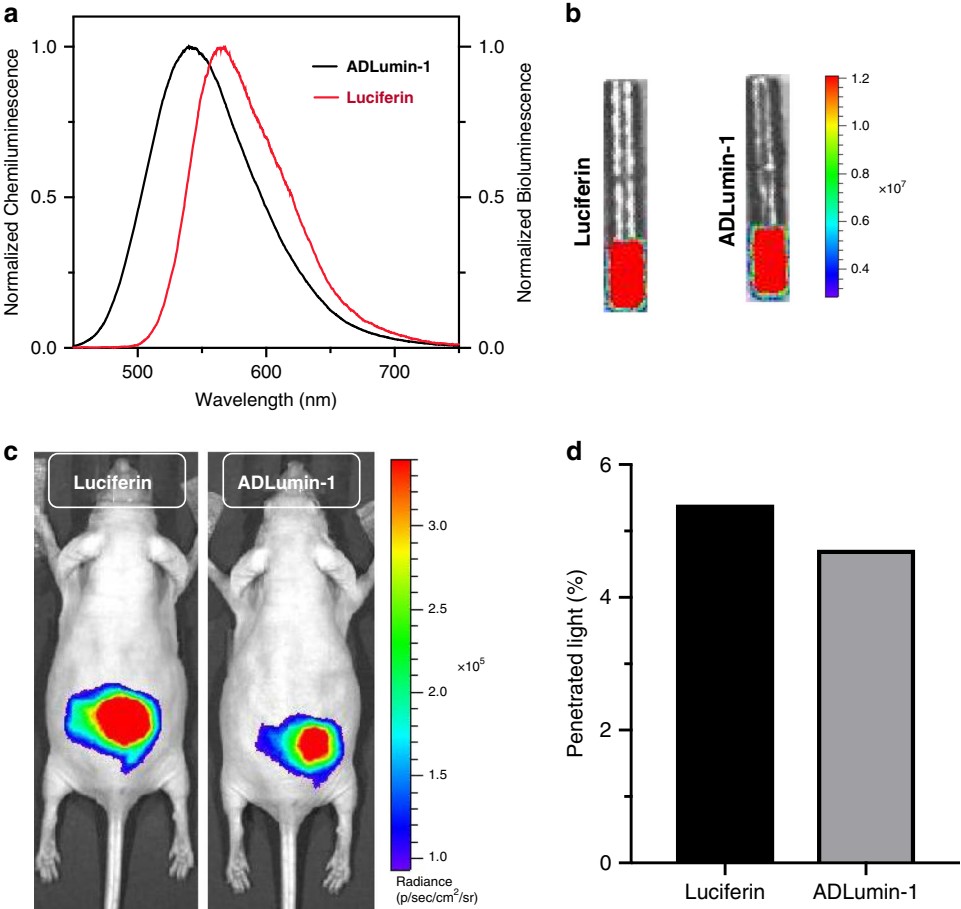

**Fig. 4 Chemiluminescence light penetration of ADLumin-1. a** Chemiluminescence spectra of ADLumin-1 (black) and bioluminescence spectra of Luciferin (red); **b** Light penetration study with Luciferin and ADLumin-1. A glass tube filled with a solution of firefly luciferin with luciferase or ADLumin-1 with similar intensities before putting under the abdomen of two nude mice; **c** Images that were captured from the dorsal side with an IVIS imaging system after placing the tubes; **d** The quantification of the penetrated light in **c**. Nearly 5% light penetrated the whole body in both groups.

were 1.7-fold and 1.8-fold at 15- and 30-min, respectively (Fig. 6a). To further confirm the signals originated from brain, we also conducted imaging with multiple filters from 500 to 840 nm (Supplementary Fig. 6A). We found that the highest signals were from the 640 nm filter, with which the difference with 640 nm was 1.7-fold. This is similar to the open filter setting. These results suggested that the majority of the signals are from brains (640 nm), but not from shallow locations such as skin.

Interestingly, we also noticed that the chemiluminescence signals from eyes were higher (1.6-fold) in the 5xFAD group than that in WT group (Fig. 6c, d). This is consistent with our previous report, in which we showed that NIRF probe CRANAD-X (X = −2, −3, −30, −58, and −102) could detect the Aβ content in eyes[56]. Even more intriguingly, we found that the signals from noses were also stronger in the 5xFAD group (1.7-fold, Supplementary Fig. 6B). No significant difference was observed from the palms of the WT and 5xFAD groups (Supplementary Fig. 6D).

**Dual-amplification of signal via chemiluminescence resonance energy transfer (DAS-CRET).** Although we demonstrated that ADLumin-1 was a smart chemiluminescence probe for Aβs via testing in pure solutions, brain homogenate, and in vivo brain and eye imaging, its emission peak was still relatively short. To overcome this problem, we speculated that CRET could be used to shift the detectable emission into the near infrared (NIR)

window. Since CRET does not need external excitation and has no interference from the cross-excitation of the acceptor[57–59], data analysis is easier. More importantly, better penetration could be achieved, due to the longer emission of the acceptor. The requirements for CRET are similar to FRET[60], which requires that the donor and acceptor are conjugated through a linker or proteins engineered in close proximity (normally <10 nm). The closer proximity of the donor-acceptor and more spectral overlap of donor emission and acceptor excitation, the higher the transfer efficiency.

Interestingly, our previous work demonstrated that non-conjugated FRET was feasible in solutions with two small molecules that both have binding capacity to Aβ species, because the pair have a high probability of randomly incorporating themselves into the beta-sheets of Aβs within 10 nm proximity to generate a viable FRET signal[61]. However, this technology was not very efficient and nearly impossible to apply in vivo, due to its short excitation and cross-excitation of the acceptor, which makes it complex and hard to tease out the FRET effect. Although CRET has been used for various biological studies and in vivo imaging, most of CRET pairs have no signal amplification (turn-on) for either donor or acceptor upon target binding. Here we hypothesized that dual-amplification of signal-CRET (DAS-CRET) could be feasible if both the donor and acceptor can bind to the target and amplify (turn-on) their luminescence signals (Fig. 7a). In this report, we used the nonconjugated pair of ADLumin-1 and CRANAD-

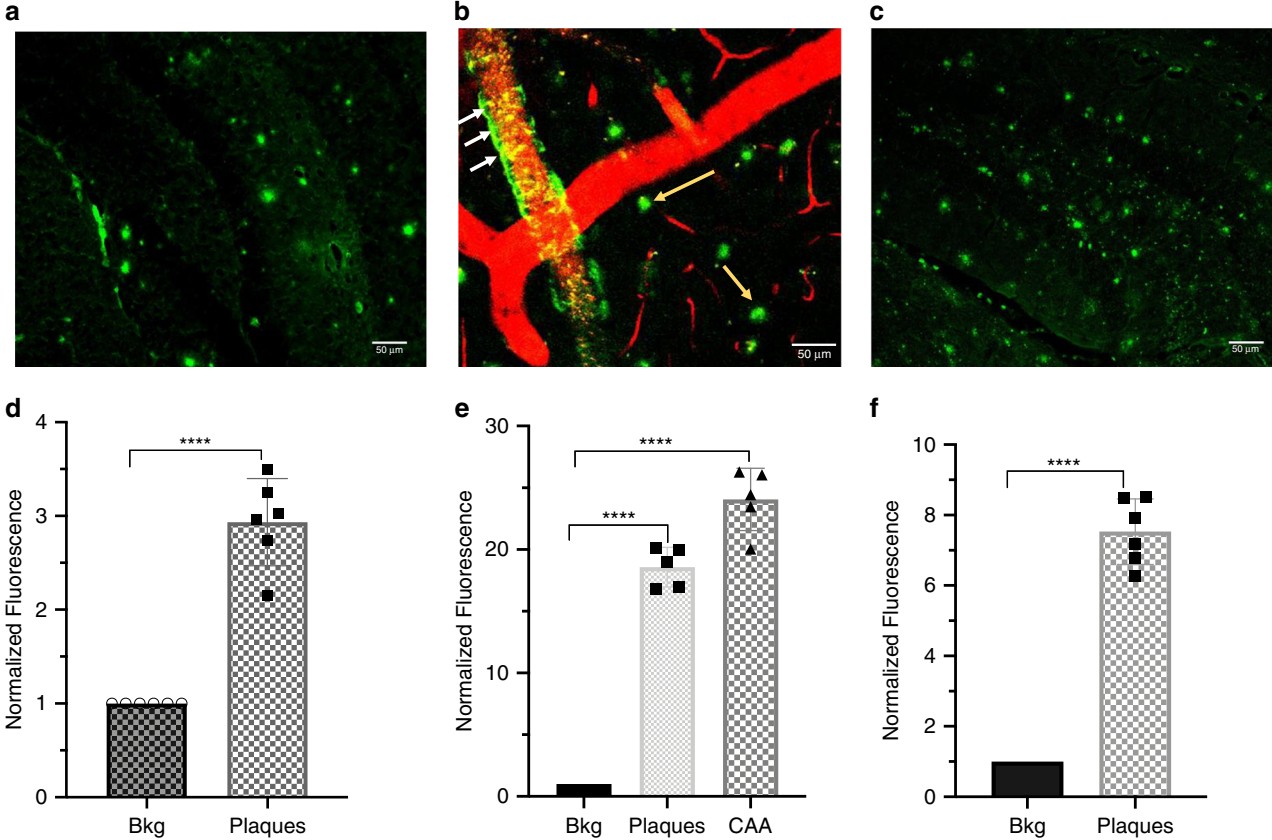

**Fig. 5 In vitro and in vivo plaques imaging with ADLumin-1. a** Representative brain slice image of cortex area of a 24-mo-old APP/PS1 mouse with ADLumin-1 staining; **b** Two-photon microscopic image of brain cortex area with ADLumin-1 in an alive 15-mo-old 5xFAD mouse. The blood vessels were highlighted by Rhodamine B isothiocyanate-Dextran, and the presentative image is merged images from two concurrent imaging channels. The yellow arrows indicate plaques, and the white arrows indicate CAA labeling; Scale bar: 50 μm. **c** Ex vivo histology of a mouse brain slice obtained after the two-photon imaging with ADLumin-1. **d–f** Quantitative analysis of SNR of plaques of in vitro slice (**d**) (mean ± s.d., $n = 6$ independent measurements), in vivo plaques (mean ± s.d., $n = 5$ independent measurements) and CAAs (mean ± s.d., $n = 5$ independent measurements) (**e**), and ex vivo plaques (mean ± s.d., $n = 6$ independent measurements) (**f**). $P < 0.0001$ (**d**); $P < 0.0001$ (**e**); and $P < 0.0001$ (**f**). The $P$-values were generated by Graphpad Prism 8 with two-tailed unpaired $t$-test; ***$P$-value $< 0.001$, ****$P$-value $< 0.0001$. *Bkg* background, *CAA* cerebral amyloid angiopathy. Source data underlying **d–f** are provided as Source Data file.

$3^{30}$, a smart NIRF probe for Aβs, to demonstrate that DAS-CRET was achievable with Aβs in solutions, in brain homogenates, and in vivo whole brain imaging. We believe this strategy can also be extended to detect other misfolding-prone proteins.

We first tested whether DAS-CRET was possible in PBS solutions. Previously, we reported that CRANAD-3 was a turn-on NIRF probe for Aβs in vitro and in vivo[30]. In this test, we incubated ADLumin-1, CRANAD-3, and Aβ aggregates in PBS buffer, and found that CRET between ADLumin-1 and CRANAD-3 was very obvious in the presence of Aβs, while there is no observable CRET in the absence of aggregates (Fig. 7b). Remarkably, the CRET was not only FRETing into longer emission, but it also significantly amplified the signals. Compared to the CRET pair without Aβs, the signal was about a 133-fold increase with the presence of Aβs at 660 nm (Fig. 7b). This large amplification is due to dual signal amplification that arose from turn-on effects from both ADLumin-1 and CRANAD-3 in the presence of Aβs. To the best of our knowledge, this phenomenon is exceptional for CRET. This highly efficient CRET is also likely due to the very close proximity of the pair when they randomly inserted into beta-sheets of Aβ aggregates, which are consisted of abundant fibrils that contains numerous beta-sheets in parallel or anti-parallel arrangement. In addition,

the well-overlapped spectra of ADLumin-1 emission and CRANAD-3 excitation also contributed to the highly efficient CRET (Supplementary Fig. 6C).

To further investigate whether this CRET is feasible in a brain-like environment, we conducted similar experiments with mouse brain homogenate. Indeed, we observed an apparent CRET signal at 640 nm (Fig. 7c), which is corresponding to the emission of CRANAD-3. The CRET signal was increased 11.4-fold if Aβs were added to the homogenate. Remarkably, we found that the signals from DAS-CRET were 2.38-fold higher than that from the non-CRET group (without CRANAD-3) at 640 nm, suggesting that DAS-CRET is achievable in a real biologically relevant environment.

To accurately analyze the signal amplification, FRET is also coupled with spectral unmixing technique to separate the signal contributions from the donor and the acceptor[62,63]. Although spectral unmixing has been rarely explored for CRET, we speculated that spectral unmixing could be feasible for our DAS-CRET. Indeed, the detection sensitivity could be further increased via spectral unmixing. In Fig. 7d, it is very clear that the unmixed spectrum (red line) is similar to the emission of CRANAD-3 after unmixing. Remarkably, the difference between CRET and non-CRET could reach 31-fold in the brain-like environment.

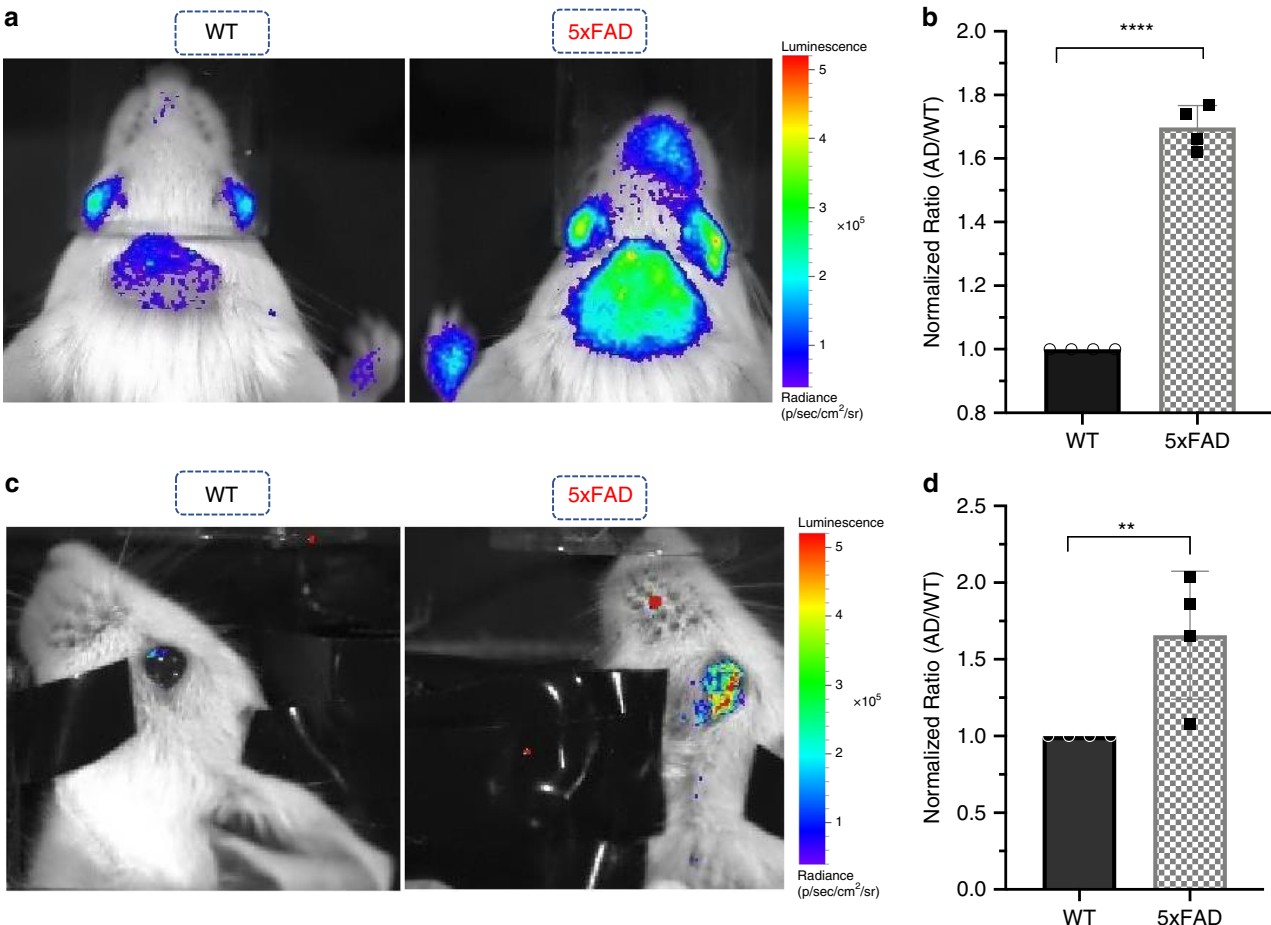

**Fig. 6 In vivo chemiluminescence imaging with ADLumin-1. a** In vivo brain imaging of WT and 5XFAD mice with iv injection of ADLumin-1; **b** Quantitative analysis of the images in **a**; **c** Representative images of eyes in WT and 5xFAD mice after iv injection with ADLumin-1; **d** Quantitative analysis of the images in **c** ($n = 4$). Error bar: Stdev; Data points with its error bar stands for mean ± s.d. derived from $n = 4$ biologically independent animals. $P < 0.0001$ (**b**); $P = 0.0098$ (**d**). The $P$-values were generated by Graphpad Prism 8 with two-tailed unpaired $t$-test; ***$P$-value < 0.001; **$P$-value < 0.01. AD Alzheimer's disease, WT wild-type. Source data underlying **b** and **d** are provided as Source Data file.

**In vivo DAS-CRET imaging**. To explore whether the CRET pair can be used in alive animals, we first mimicked the in vivo environment via subcutaneously injecting the mixture of the CRET pair with Aβs into a nude mouse at the ventral hind limb, and ADLumin-1 + Aβs as the control group (Fig. 8a). At 15 min post injection, the signal from the CRET group (right) was 1.34-fold higher than that from the control site (left) (Fig. 8b), suggesting that the nonconjugated CRET pair can be used in vivo. To investigate whether the CRET pair can be used for deep locations, we flipped the above mouse to acquire images from the dorsal side (the thickness from the injection site to the dorsal surface is about 1.2 cm). As expected, signals could be detected. Remarkably, we found the difference between the CRET group and the control group could be easily observed (1.42-fold), indicating the pair can be used for imaging at deep locations (Fig. 8b). To further achieve more accurate information about the efficiency of CRET in vivo, we conducted spectral unmixing imaging with 18 filters to collect signal from 500 to 840 nm. The unmixed spectra closely resemble the spectra of ADLumin-1 + Aβs (green), and CRET + Aβs (red), respectively (Fig. 8c, d). After spectral unmixing, the difference between the control and the CRET groups reached 2.72-fold (Fig. 8c middle). From the composite image, it is clear that the signal from the CRET site is from long emission, while the signal from the control site is from short emission (Fig. 8c right). Taken together, these results suggested

that in vivo DAS-CRET was feasible and spectral unmixing imaging was applicable for in vivo imaging, and larger margin of differences can be achieved via spectral unmixing.

To validate the feasibility of CRET in vivo brain imaging, we injected (iv) a solution containing both ADLumin-1 and CRANAD-3, and images were collected with an open filter and 18 filters from 500 to 840 nm. With the open filter setting, we observed 2.04-fold differences between the 5xFAD group and the WT group at 15 min after the injection (Fig. 9a, c). With the 640 nm filter, the AD group showed 2.25-fold higher signal than the control WT group (Fig. 9d). Moreover, as expected, the CRET pair provided higher luminescence signals at 640 nm than that of ADLumin-1 alone in both AD and WT groups, and the increase was 2.10- and 1.66-fold, respectively (Fig. 9b). To investigate whether spectral unmixing can be applied to this in vivo imaging study, we conducted spectral unmixing with the previously described method. Indeed, the margin between AD and WT was significantly increased after spectral unmixing, and it reached 3.22-fold from the CRET pair (Fig. 9e).

In our previous studies, we reported that NIRF ocular imaging could be used to differentiate AD and WT mice and monitor therapeutic effect on Aβ-lowering with treatment[56]. In this work, we performed ocular CRET imaging as well, and found that it was feasible and provided a considerably large margin of 2.11-fold between 5xFAD and WT mice (Supplementary Fig. 7). This large

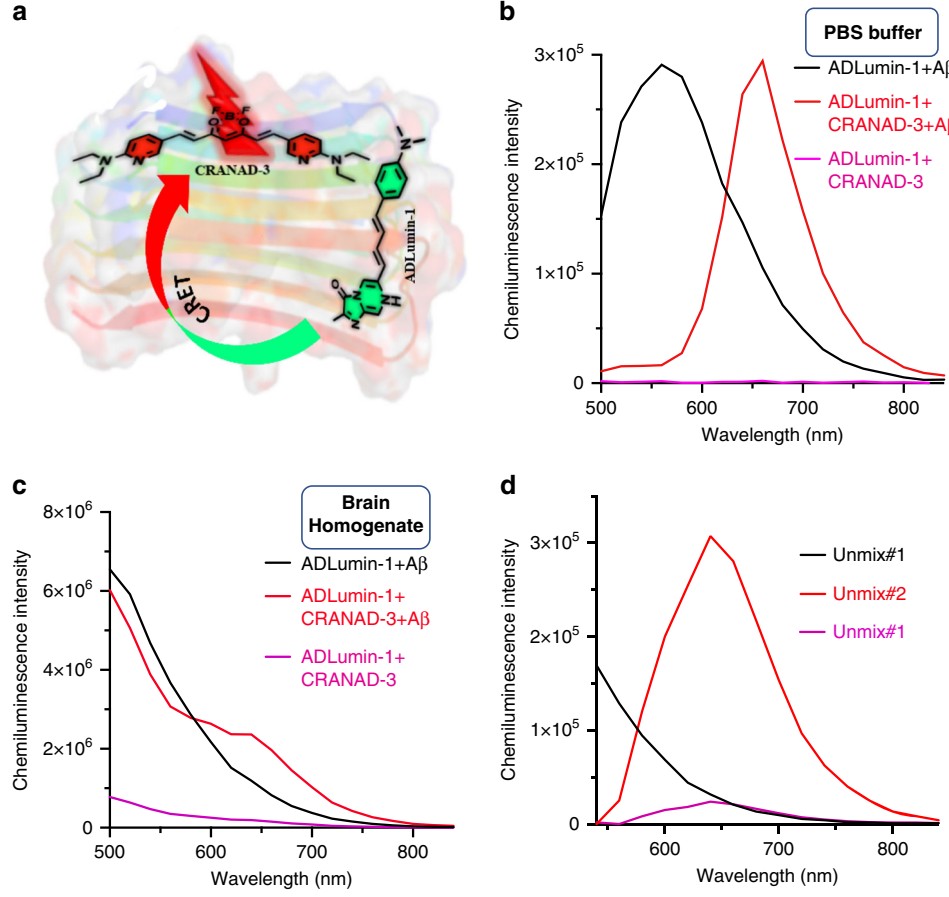

**Fig. 7 In vitro validation of CRET with ADLumin-1/CRANAD-3 pair. a** Proposed CRET model between ADLumin-1 and CRANAD-3 with Aβ40 fibrils. Two nonconjugated molecules upon binding to fibrils to bring ADLumin-1 (Donor) close enough to CRANAD-3 (Acceptor) to initiate CRET; **b** Spectrum of the CRET pair with Aβ40 fibrils in PBS (red line), and the peak was consistent with the emission of CRANAD-3 in the presence of Aβ40 fibrils. Chemiluminescence spectrum of ADLumin-1 with Aβ40 fibrils (black line); and spectrum of the mixture of ADLumin-1 and CRANAD-3 without Aβ40 fibrils (pink line). The FRETing efficiency was very high, evident by the low intensity at 500–560 nm range; **c** Spectrum of the CRET pair in brain homogenate (red line), evident by a decrease in ADLumin-1 emission (black line) and increase in CRANAD-3 emission; **d** Spectral unmixing of DAS-CRET to separate the contribution from ADLumin-1 only (Unmixed #1), and CRET (Unmixed #2) and CRANAD-3 only (Unmixed #3).

margin indicated that ocular CRET could be a very useful tool for monitoring the changes in Aβ concentrations. Interestingly, we also noticed that the nose signal was higher from the AD group than that from the WT group with ADLumin-1 alone or with DAS-CRET (Supplementary Fig. 7D).

## Discussion

In this report, we demonstrated that, ADLumin-1 is a smart chemiluminescence probe for Aβs in vitro and in vivo. We also revealed that DAS-CRET was feasible via a nonconjugated dual-turn-on CRET pair with the combination of a smart chemiluminescence probe and a smart NIRF probe. Although we only performed validations with Aβ species, we believe that the strategy for probe designing and CRET method can be extended to other misfolding proteins such as tau, alpha-synucleins, TDP-43, amylin, fibrinogen, prion, fused in sarcoma (FUS) protein, SOD, and transthyretin[31,64]. These proteins probably also meet the designing requirements, which include hydrophobic beta-sheets for binding of specific small molecule probes to turn-on luminescence, and the close proximity of abundant beta-sheets to randomly positioning the paired nonconjugated probes for CRET occurrence. It is known that not all amyloid proteins exist in brains, some misfolded protein deposits can be found in the skin and other places. Considering that our method has relatively deep

penetration as we demonstrated in the mimicked imaging (Fig. 8), our CRET strategy is likely applicable for this purpose.

For most hydrophobic fluorescence probes (for nonactivatable probes), the restricted conformation of a probe and the hydrophobicity of the residing environment of the probe have strong influence on the turn-on effect of fluorescence[31,43,65–67]. Similarly, this principle can be applied to chemiluminescence probes. ADLumin-1, which is hydrophobic, showed low efficiency of chemiluminescence in the pure solutions (10% DMSO) without Aβ aggregates, likely because of the free rotation of the carbon–carbon bond in ADLumin-1. In contrast, upon binding to Aβ fibrils, the rotation is rigidly restricted to give rigid conformations, and this conformational rigidifying could significantly enhance the chemiluminescence signal. In addition, like smart fluorescence probes, the chemiluminescence efficiency can be significantly enhanced if the probe binds to hydrophobic micro-environment. This is exactly the case for the binding of ADLumin-1 and Aβ fibrils. The observed substantial blue-shift of emission of the probe with Aβs in the fluorescence studies (Supplementary Fig. 2A), and 2) supported that ADLumin-1 was binding to a hydrophobic environment, which was further agreed by molecular docking results, which revealed that ADLumin-1 preferred to bind to the hydrophobic groove formed by Phe19, Ala21, Val24, Asn27, and Ile31 (Fig. 3c).

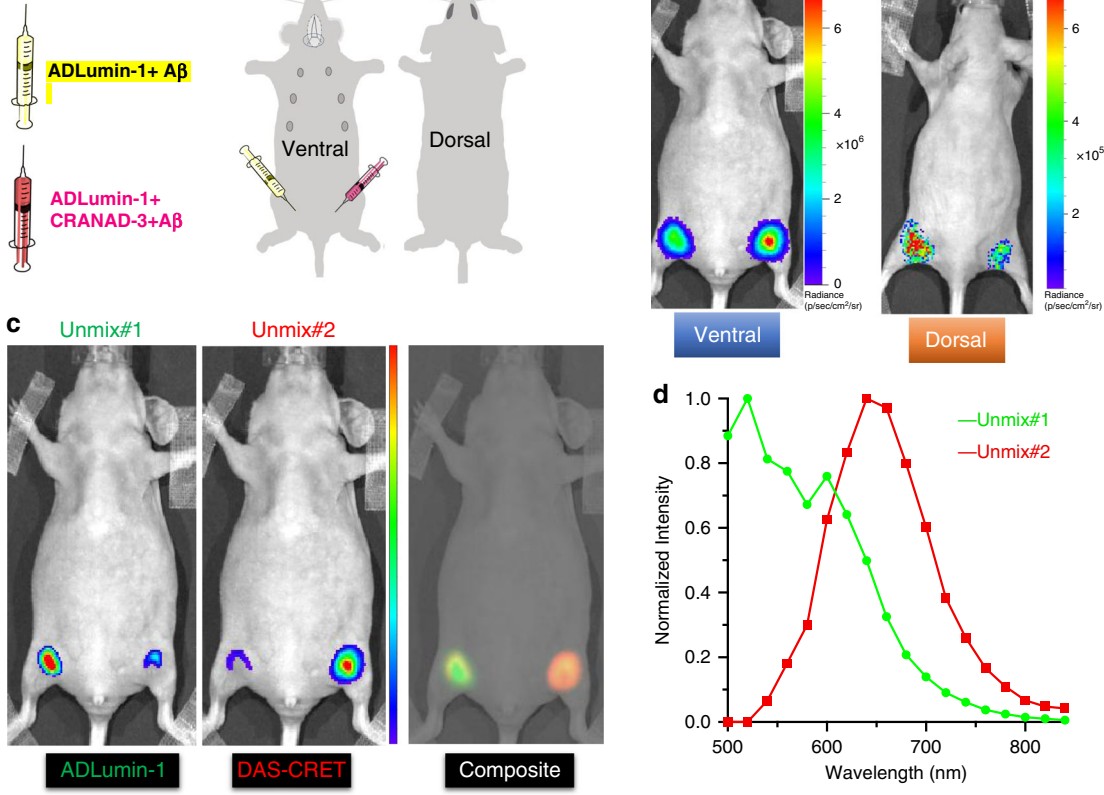

**Fig. 8 Observation of CRET under in vivo mimic conditions. a** A mixture of ADLumin-1 and CRANAD-3 with Aβ40 aggregates (CRET + Aβ) were injected subcutaneously into the right inner thigh of a female nude mouse and the control group of ADLumin-1+Aβ was injected into the left side; **b** The mouse was imaged from both the ventral and dorsal sides; **c** Spectral unmixing was conducted with sequence imaging at 15 min post injection to separate signals from ADLumin-1 (left, green) and from CRET pair (middle, red), and composite image of the two unmixed components (right); D Spectra of the unmixed contributors, ADLumin-1+Aβ (green), and CRET + Aβ (red).

Several surprising results were discovered from our studies. Although ADLumin-1 has a moderately short chemiluminescence emission peak, it was an astonishment that ADLumin-1 provided acceptable tissue penetration (5–7% light penetration for the whole body of a mouse (Figs. 4c, d, and 8)) and could be used for in vivo mouse brain imaging. This is probably due to its high SNR (>2000) (Supplementary Fig. 8) and its large FWHM. In addition, the larger FWHM could also provide larger spectral overlap between the donor and the acceptor for CRET. We believe that broadening the FWHM could be utilized for designing chemiluminescence probes in the future. Our results also suggested that, at the same emission wavelength, chemiluminescence imaging could have much better tissue penetration, which is consistent with a previous report that claimed 4-cm tissue penetration could be achieved with chemiluminescence imaging at 800 nm emission[7].

Recently, great progression has been achieved for developing new chemiluminescence probes and their application. Most of chemiluminescence probes are dependent on ROS reactions and most of the probes have been used to detect ROS in solutions, cells, and in vivo[8–14]. Surprisingly, we found that ADLumin-1 was much more sensitive to $O_2$ (27-fold-change), and the responses were very quick (within seconds). Compared to the responses to $O_2$, the enhancement of ADLumin-1 signal to ROS is relatively small. Our studies suggested that the auto-oxidation of ADLumin-1 was related to $O_2$ levels, and this oxidation could be utilized for in vivo imaging. The mechanism of auto-oxidation has not been well studied. Although full elucidation of the mechanism needs more intensive investigation, we believe that

ADLumin-1 and its analogues could be used as $O_2$ sensors in future studies.

Chemiluminescence has been used in many other fields such as cancers, diabetes, and infectious diseases. However, to the best of our knowledge, none has been reported for brain disorders including AD. It is also quite challenging to apply CRET for brain imaging. Fortunately, we successfully demonstrated that it was feasible in vivo, as the nonconjugation of the pair could be an advantage to avoid high molecular weight for probe designing. We believe that the nonconjugation strategy can be easily adapted by other groups for other misfolding proteins.

Signal amplification is always beneficial for imaging. In this report, we showed that the chemiluminescence of ADLumin-1 could be amplified (turned-on) by Aβ species. We further validated that the emission of ADLumin-1 could be pushed into the NIR window via CRET. Our images showed impressive signal to noise ratio (SNR). This is likely due to no excitation leakage and autofluorescence. Since it is difficult to have deep penetration from short chemiluminescence, the signal detected at 640 nm is primarily from CRET. In addition, we demonstrated that spectral unmixing can significantly improve the detection sensitivity both in vitro and in vivo for CRET imaging.

Previously, we demonstrated that NIRF probes CRANAD-X could be used to image Aβ contents in eyes of AD mice[56]. Consistent with this report, here we showed that higher luminescence could be observed from the eyes of AD mice via ADLumin-1 alone or via DAS-CRET. Further investigation of eye imaging is still undergoing in our group. The higher chemiluminescence signal from AD noses is also quite intriguing. A few

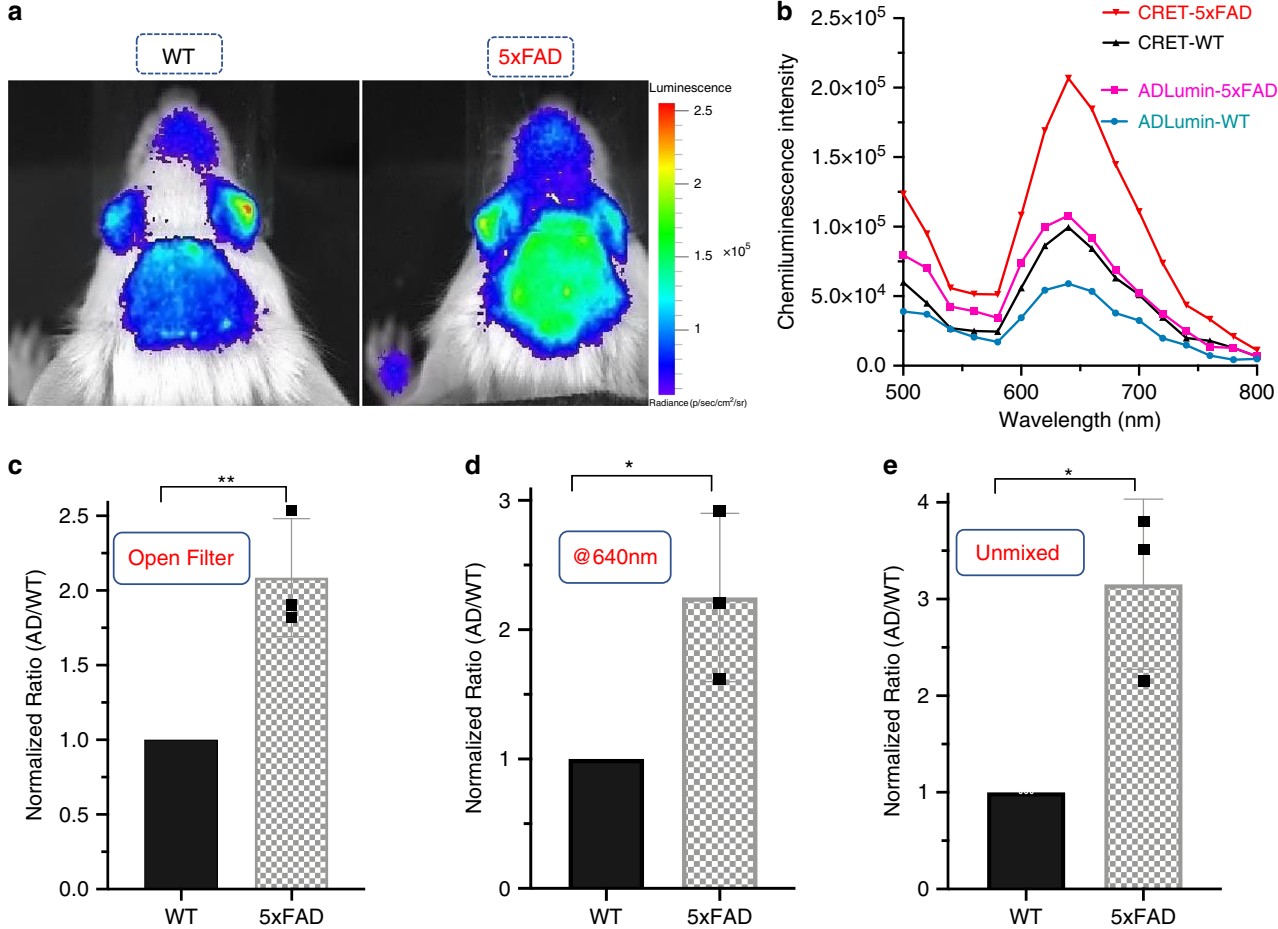

**Fig. 9 In vivo DAS-CRET imaging. a** In vivo brain imaging of WT and 5XFAD mice with mixture of ADLumin-1 and CRANAD-3; **b** In vivo emission spectra of CRET pair (red line-5xFAD) and control (black line-WT), and ADLumin-1 only (pink line-5xFAD) and ADLumin-1 only control (blue line-WT); **c–e** Quantitative analysis of the images with CRET pair with the setting of open filter (**c**), at 640 nm (**d**) and after spectral unmixing (**e**). Data are represented as mean ± s.d. with $n = 3$ biologically independent animals. $P = 0.0059$ (**c**), $P = 0.0292$ (**d**), $P = 0.0133$ (**e**). The P-values were generated by Graphpad Prism 8 with two-tailed unpaired t-test; *P-value < 0.05, **P-value < 0.01. Source data underlying **c–e** are provided as Source Data file.

reports have claimed that noses in AD patients have higher Aβ contents[68,69]. Further validation in patients and AD mice is still needed.

Normally, to analyze the Aβ contents from brain tissue homogenate or CSF, certain processes are needed after homogenization. The post-homogenization processes include extraction with SDS, TBS and/or formic acid, and centrifugation to separate supernatants and pellets, which are necessary for ELISA analysis, the most used method for reporting Aβ concentrations. It is well-known that these steps strongly contribute to the poor reproducibility of ELISA. In this report, our in vitro experimental results showed that ADLumin-1 could detect the presence of Aβ species in brain homogenate, indicating that ADLumin-1 could be used for reporting Aβ concentrations without any processing. The optimization is still ongoing, and we will report the results in due course.

In summary, we reported that ADLumin-1 was a smart chemiluminescence probe for Aβs in vitro and in vivo, and DAS-CRET was feasible for detecting Aβ contents in the NIR window. Considering the tissue penetration of DAS-CRET, it is possible to use our method for brain studies on large animals such as marmosets. In addition, our results also showed that the Aβ contents in eyes could be detected. We believe that our methods have strong potential for preclinical animal studies and could be possibly applied for translational studies in the future.

## Methods

**Oxygen and ROS sensitivity of ADLumin-1.** A 100.0 uL DMSO solution of ADLumin-1 (200.0 μM) was added to a well in a 96-well plate. One group was bubbled with oxygen for 5 s at different time points, and the control group was sealed without bubbling. Once the bubbling was over, the plate was subjected to chemiluminescence intensity recording with the IVIS system with the open filter setting (500–840 nm). For ROS tests, A PBS solution (240 μL) was incubated in an eppendorf tube. To each of above tubes, different ROS (250.0 μM, $H_2O_2$ 30 μL, $ClO_4^-$ 30 μL, TBHP 30 μL, and $KO_2$ 30 μL) was added, respectively. Then, a solution of ADLumin-1 in DMSO (30 μL, 250.0 μM) was added. Finally, 100.0 uL of the resulting solution was transferred into a well of 96-well plate, and triplicated samples were prepared. Chemiluminescence images were obtained with an IVIS system under the open filter (500–840 nm).

**Preparation of Aβ40 aggregates.** 1.0 mg of Aβ40 peptide (TFA) was suspended in 1% ammonia hydroxyl solution (1.0 mL). Then 100 μL of the resulting solution was diluted 10-fold with PBS buffer (pH 7.4) and stirred at room temperature for 3 days. Transmission electron microscopy and Thioflavin T solution test were used to confirm the formation of aggregates[27].

**In vitro fluorescence spectra of ADLumin-1 with Aβ40 aggregates.** To test the interactions of ADLumin-1 with Aβs, we used the following three-step procedure. In step 1, 1.0 mL of double-distilled water was added to a quartz cuvette as a blank control, and its fluorescence was recorded with the same parameters used for ADLumin-1. In step 2, the fluorescence of an ADLumin-1 solution (1.0 mL, 250.0 nM) was recorded with excitation at 410 nm and emission from 450–700 nm. In step 3, to the above ADLumin-1 solution, 10.0 μL of Aβs (25.0 μM stock solution in PBS buffer for Aβ40 aggregates) were added to make the final Aβ concentration of 250.0 nM. Fluorescence readings from this solution were recorded as described in

step 2. A blank control from step 1 was used to correct the final spectra from steps 2 and 3.

**In vitro chemiluminescence study**. A PBS solution (95 μL) or brain homogenate (95 μL) was incubated in an eppendorf tube in the presence or absence of Aβ aggregates (final concentration 12.5 μM). To each of the above tubes, a solution of ADLumin-1 in DMSO (5.0 μL, 250 μM) was added. For the CRET test, 5.0 μL DMSO solution of CRANAD-3 (250.0 μM) and 5.0 μL DMSO solution of ADLumin-1 (250.0 μM) were added (Aβ aggregates final concentration 12.5 μM). Finally, 100.0 uL of the resulting solution was transferred into a well of 96-well plate, and triplicated samples were prepared. Chemiluminescence images were obtained with an IVIS system under the open filter (500–840 nm) or specific filters.

**Binding affinity of ADLumin-1 with Aβs**. A series of solutions containing 250.0 nM of ADLumin-1 and various concentrations of Aβ40 aggregates (0.0, 25.0, 100.0, 250.0, 500.0, 1000.0, 2000.0, 4000.0, and 8000.0 nM) were subjected to fluorescence spectral recording (Ex = 420 nm, Em = 450–700 nm). The emission readings at 515 nm were used for a nonlinear specific binding fitting.

**In vitro histological study**. A fresh brain tissue from a 24-month old APP/PS1 mouse was fixed in 4% formaldehyde for 24 h and transferred into 30% sucrose at 4 °C until the tissue sunk. Then the tissue was embedded in OCT with gradual cooling over dry ice. The OCT embedded tissue block was sectioned into 25-μm slice with a cryostat. 25 μM of ADLumin-1 in 50% ethanol/PBS was prepared as the staining solution. The brain slices were incubated with freshly prepared staining solution for 15 min at room temperature and then washed with 70% ethanol for 1 min, 50% ethanol for 1 min, followed by washing with double-distilled water twice. Then the slice was covered with FluoroShield mounting medium (Abcam) and sealed with nail polish. Florescence images were obtained using the Nikon Eclipse 50i microscope with a blue light excitation channel.

**In vivo two-photon imaging**. A 15-month-old 5xFAD female mouse was anesthetized with 2% isoflurane, and a cranial imaging window was surgically prepared as described[30,70]. Before ADLumin-1 injection, two-photon images of capillary were acquired using 900-nm laser (Prairie Ultima) with 570–620 nm emission by injection of Rhodamine B isothiocyanate-Dextran. A bolus i.v. injection of ADLumin-1 (4 mg/kg in a fresh solution of 15% cremophor, 15% DMSO, and 70% PBS) was administered at time 0 min during image acquisition. The images were acquired with an emission channel of 500–550 nm. For imaging, we used a two-photon microscope (Olympus BX-51) equipped with a ×20 water-immersion objective (0.45 numerical aperture; Olympus)[30,70]. Single Images were collected with 512 × 512 pixel resolution. The T-Series images were acquired for 20 min continuously for the same regions. Image analysis was performed with ImageJ software.

**Ex vivo histological study**. The mouse used for two-photon imaging with ADLumin-1 was sacrificed at 2 h after the injection. The fresh brain tissue was fixed in 4% formaldehyde for 24 h and transferred into 30% sucrose at 4 °C until the tissue sunk. Then the tissue was embedded in an OCT block, which was sectioned into 25-μm slice with a cryostat. Then the slice was covered with FluoroShield mounting medium (Abcam). Florescence images were observed with a Nikon Eclipse 50i microscope.

**In vivo mimic demonstration of DAS-CRET**. Two different PBS solutions were prepared for this test. The first PBS solution is consisted of 5 μL of ADLumin-1 (250 μM) and 45 μL of the Aβ40 Aggregates (25 μM). The second solution is consisted of 2.5 μL of ADLumin-1 (500 μM), 2.5 μL of CRANAD-3 (500 μM) and 45 μL of the Aβ40 Aggregates (25 μM). After solutions were prepared, one female 8-month-old nude mouse was anesthetized, and the left hind limb was subcutaneously injected with the first solution and the right hind limbs was subcutaneously injected with the second solution. The mouse was then placed into the IVIS imaging chamber and images were captured from the dorsal side and from the ventral side with the open filter (500–840 nm) and specific filters.

**In vivo chemiluminescence and DAS-CRET imaging**. Five-month-old female 5xFAD mice (n = 4) and age-matched female wild-type control mice (n = 4) were shaved before background imaging and were intravenously injected with freshly prepared ADLumin-1 (4 mg/kg, 15% DMSO, 15% cremophor, and 70% PBS) or the mixture solution of ADLumin-1 and CRANAD-3 (the dose for both of the dye was 4 mg/kg with the formulation of 15% DMSO, 15% cremophor, and 70% PBS). The IVIS Spectrum animal imaging system (PerkinElmer) was used for in vivo imaging. Images were acquired with open filter or specific emission filters. Living Image 4.2.1 software (PerkinElmer) was used for data analysis. Chemiluminescent signals from the brain were recorded before and 5, 15, 30, 60 min after i.v. injection of the probe. Chemiluminescent signals from the eyes were recorded at 0 and 45 min after i.v. injection of the probe. Sequence images were captured at 15 min after

probe injection with the following parameters: sequence filter from 500 to 840 nm with an interval of 20 nm; Exposure time is 30 s, FOV = D. Spectral unmixing was performed with Living Image 4.2.1 software, and guided unmixing method was selected. To evaluate our imaging results, an ROI was drawn around the brain or the eye region. Student t-test was used to calculate P-values.

**Reporting summary**. Further information on research design is available in the Nature Research Reporting Summary linked to this article.

## Data availability
All data supporting the findings of this study are included in the Article and its Supplementary Information. Source data are provided with this paper.

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

## Acknowledgements

This work was supported by NIH R01AG055413, and R21AG059134 awards (C.R.). We thank Pamela Pantazopoulos, B.S. for proofreading this manuscript.

## Author contributions

C.R. designed the project. J.Y. and W.Y. Performed in vitro experiments and in vivo imaging. R.V. and Y.S. contributed to docking studies. W.Y. synthesized and characterized the compounds. K.Y., P.W., C.Z., and M.K. contributed to in vitro spectral studies and provided suggestions for experiments, B.Z. and K.R., contributed to tissue staining. C.Z. provided reagents and suggestions for experiments. C.R., J.Y., and W.Y. analyzed the results, prepared the manuscript, figures and supplementary information. All authors contributed to discussion and editing of the manuscript.

## Competing interests

The authors declare no competing interests.
