## [Peer Review File · Nature Communications]

REVIEWER COMMENTS

Reviewer #1 (Remarks to the Author):

In this work, the authors reported about the design and development of a new chemiluminescent probe (ADLumin-1) that is able to "turn on" its chemiluminescence in presence of the target analyte (A β) and they tested this approach both in vitro and in vivo. Moreover, to overcome the limitation of short emission of ADLumin-1, they demonstrated the feasibility to achieve dual-amplification of signal via chemiluminescence resonance energy transfer (DAS-CRET) with two non-conjugated smart probes in solutions, tissues and brain homogenates, and in vivo whole brain imaging.

This work can be of great interest for different sectors: for imaging-based diagnostics but also for the synthesis of a new generation of chemiluminescent tracers that activate themselves in the presence of the target analyte, making them independent of the addition of other reagents, paving the way for a reagentless approach. This may represent a turning point in the field of the use of chemiluminescence as an ideal optical detection method which thus becomes highly competitive with colorimetric or fluorescent based methods. Indeed, although ADLumin-1 was designed for A β s, the authors believe that this strategy could be potentially extended to a wide range of targets.

The discussion is very thorough, the statistics of the mice used for the experiments is not very extensive but can be used as proof of concept. It is a very large work, which includes the synthesis of the chemiluminescent marker, its photophysical characterization and in vitro and in vivo tests. The steps are well detailed and exposed in a logical and orderly way.

The only thing that in my opinion deserves a deepening is the topic of in vivo toxicity of these molecules. Biocompatibility is indeed a crucial aspect in the design and development of markers to be used in vivo.

Reviewer #2 (Remarks to the Author):

This paper describes the evaluation of two (mostly one) new chemiluminescent compounds that putatively bind to A β species and exhibit enhanced emission of light when bound compared to free in solution. The general idea is novel and the results represent a nice demonstration of the power of this approach for labeling of amyloids for optical detection in vitro and in vivo. While the results presented are fairly convincing, the manuscript could be improved significantly by inclusion of more thorough characterization of the compound-amyloid complex, increased discussion regarding the mechanism for enhanced luminescence, and general improvements in clarity of experiments and data presented.

Overall, it is a very nice story and should be publishable after some careful revisions.

The following are a few specific comments that the authors should consider:

-Should provide direct evidence of binding to A β species. While the structure looks like it should bind to fibrils, the authors should provide some data to support what types of aggregates the ADLumin-X compounds bind.

-Should provide some rationale (ideally with supporting data) for why the ADLumin-X compounds show enhanced chemiluminescence when bound to A β versus free in solution. Oxygen levels are the same whether the A β samples are present or not, so how could the presumed binding of the compounds to A β lead to enhanced chemiluminescence in the presence of O₂?

-Figure 1A, Moiety B implies that ADLumin-X intercalates between peptides within a larger beta sheet superstructure. This binding mode appears different than the computed binding mode shown in Figure 3C? The authors should reconcile this discrepancy and remain consistent throughout the manuscript. Same issue with Figure 7A. Is the binding mode for CRANAD-3 known?

-Figure 5B: It is unclear what is being imaged here. Is this an image from a living 5xFAD mouse, as the figure caption and main text might suggest? What is the scale bar and how was this image acquired? How were the blood vessels stained? What region of the brain is being shown?

-Figure 5C: If this image is an ex vivo slice of the mouse brain used to generate the image in Figure 5B, why aren't the blood vessels shown differentially stained? What happened to the vascular plaques associated with CAA?

Reviewer #3 (Remarks to the Author):

In this manuscript, the authors reported a smart chemiluminescence probe for Abs in vitro and in vivo, in order to realize the NIR imaging, DAS-CRET was carried out for the detection of Abs (in the brain and eyes) in living 5XFAD mice. It's certainly an interesting paper, but the following questions should be further clarified before acceptance in Nature Communications or other journals.

1. ADLumin-1 can bind to Abs, but with lower affinity ($K_d = 2.1 \mu\text{M}$) compared with other Ab probes. In addition, according to the author's description and experimental results, ADLumin-1 was very sensitive to oxygen, and quickly (within seconds) converted to ADLumin-3, and how is its affinity to Abs?
2. In figure 3B, why the chemiluminescence spectra of ADLumin-1 alone is very low? After reacting with DMSO, there should be some light released, and in addition, the chemiluminescence was dramatically turned-on with A β 40 aggregates, that's hard to understand, without the excitation light source, the energy is conserved, and the signal intensity should only depend on the concentration of ADLumin-1 and oxygen.
3. Compared with figure 3D and F, what is the major difference of oxygen or DMSO between PBS and mouse brain homogenate?
4. In vivo chemiluminescence imaging with ADLumin-1 in figure 6A and figure 9A, in addition to the signal from the nose, the palm signal of 5XFAD mice seems to have a higher intensity than the WT group, quantitative data of these areas should be carefully analyzed to exclude the influence of administration dose.
5. In figure 9A, the eye signal of WT mice seems to have a higher intensity than 5XFAD, which is contrary to the result in figure 6A.

Thank you for reviewing our manuscript titled “**Smart chemiluminescence probes and dual-amplification of signal for detection of amyloid beta species in Alzheimer’s disease model**” (NCOMMS-20-02566) by Yang, J. et al. We appreciate the opportunity to re-submit the revised manuscript. Based on your valuable suggestions and the Reviewers’ thoughtful comments, we have revised the manuscript accordingly. The changes are highlighted in red in the manuscript. We believe that in the revised manuscript, we have addressed all of the points raised by the Reviewers and have made necessary changes. The following is our point-by-point response to the Reviewers. The Reviewers’ comments are in italics.

Thank you for taking the time to consider this resubmission.

Sincerely yours,

Chongzhao Ran

Referee 1

1) The only thing that in my opinion deserves a deepening is the topic of in vivo toxicity of these molecules. Biocompatibility is indeed a crucial aspect in the design and development of markers to be used in vivo.

RE: Thank you very much for your very positive evaluation of our work. We agree that biocompatibility is very important for the probes to be used in vivo. It will be great if we can provide LD50 data; however, our lab is currently shut down due to the COVID-19 pandemic and we do not know when our lab will re-open. In the course of the validation of ADLumin-1, we have intravenously injected this compound into mice more than 20 times, and all of the mice were healthy after the injections. Here, if it is acceptable, we would like to consider this fact as evidence to support that ADLumin-1 has tolerable biocompatibility. Of course, we will provide the LD50 data if it is necessary once our lab re-opens.

Referee 2

1) Should provide direct evidence of binding to A β species. While the structure looks like it should bind to fibrils, the authors should provide some data to support what types of aggregates the ADLumin-X compounds bind.

RE: Before addressing your comments, we would like to take the opportunity to thank you for your very insightful and important critiques for our manuscript. Below are our responses to your questions .

We have taken TEM imaging after the preparation of A β 40 aggregates, and typical A β fibrils could be easily observed (SI Fig.2C). We found that there is significant fluorescence (100-fold, Fig. 3A) and chemiluminescence intensity (104-fold, Fig. 3D) increases upon interaction with A β s in PBS solution. We have also incubated ADLumin-1 with A β oligomers in PBS solution and found no significant change of chemiluminescence intensity. These results suggested that the probe had specific binding with amyloid fibrils. This was further confirmed by the docking model study shown in Fig. 3C. We have added this information into the revised manuscript (Page 7 of our manuscript).

2) Should provide some rationale (ideally with supporting data) for why the ADLumin-X compounds show enhanced chemiluminescence when bound to A β versus free in solution. Oxygen levels are the same

whether the Abeta samples are present or not, so how could the presumed binding of the compounds to Abeta lead to enhanced chemiluminescence in the presence of O₂?

RE: We thank the reviewer for raising this point. The presence of O₂ is necessary for generating chemiluminescence in solutions both with and without A β aggregates. However, in the pure solutions (5-10% DMSO) without A β aggregates, the efficiency of chemiluminescence is very low, likely because of the free rotation of the carbon-carbon bond in ADLumin-1. In contrast, in the presence of A β aggregates, the rotation will be restricted to give a rigid conformation, and this conformational rigidifying could significantly enhance the chemiluminescence signal. In addition, like “smart” fluorescence probes, the chemiluminescence efficiency can be significantly enhanced if the probe binds to a hydrophobic micro-environment. This is exactly the case for the binding of ADLumin-1 and A β fibrils. Our evidence include 1) the substantial blue-shift of emission of the probe with A β s in the fluorescence studies (SI Fig. 2A), and 2) molecular docking results further demonstrated that ADLumin-1 preferred to bind to the hydrophobic groove formed by Phe19, Ala21, Val24, Asn27, and Ile31 (Fig. 3C). We have added more detailed discussions in this revised manuscript (Page 18 of our manuscript).

3) Figure 1A, Moiety B implies that ADLumin-X intercalates between peptides within a larger beta sheet superstructure. This binding mode appears different than the computed binding mode shown in Figure 3C? The authors should reconcile this discrepancy and remain consistent throughout the manuscript. Same issue with Figure 7A. Is the binding mode for CRANAD-3 known?

RE: We thank the reviewer for pointing out this. We have made necessary changes according to your suggestions, and the new diagrams make the binding mechanism more intuitive. Based on our previous results, CRANAD-3 could interact with the core fragment KLVFF (A β 16-20) of A β s¹, suggesting CRANAD-3 and ADLumin-1 are not competitive due to their different binding models.

4) Figure 5B: It is unclear what is being imaged here. Is this an image from a living 5xFAD mouse, as the figure caption and main text might suggest? What is the scale bar and how was this image acquired? How were the blood vessels stained? What region of the brain is being shown?

RE: We thank the reviewer for pointing out these ambiguities. We have added a more detailed description into the caption of Fig. 5 in the revision. Fig. 5B is the two photon microscopic image with ADLumin-1 in the brain cortex of a living 5xFAD mouse, and the capillaries were highlighted with Rhodamine B isothiocyanate-Dextran (average MW: 10,000), which is a water-soluble polymer and can be washed out from blood within 30-60 minutes; however it is not able to bind to the vessels. The images of plaques, CAAs, and capillaries were captured concurrently with different emission collection channels.

5) Figure 5C: If this image is an ex vivo slice of the mouse brain used to generate the image in Figure 5B, why aren't the blood vessels shown differentially stained? What happened to the vascular plaques associated with CAA?

RE: For the ex vivo imaging, the brain tissue had been cut into thin sections of 25- μ m slices with a cryostat, and this sectioning process cuts the vessels into short segments. This is why the images are different for alive brain imaging and ex vivo slice imaging. It is not easy to find a whole shaped blood vessel as shown in the two-photon study. In addition, if there was any residue of the injected Rhodamine B isothiocyanate-Dextran at the time of ex vivo imaging, it had been washed away in the course of slice preparation. Regarding the CAA deposits in ex vivo imaging, it is usually hard to be identified in mouse brain slices,

due to the small sizes with hollow interiors. For human brain slices, based on our experience, sometimes it is much easier, due to the large sizes of the vessels with CAA deposits.

Referee 3

1) *ADLumin-1 can bind to Abs, but with lower affinity ($K_d = 2.1 \mu M$) compared with other Ab probes. In addition, according to the author's description and experimental results, ADLumin-1 was very sensitive to oxygen, and quickly (within seconds) converted to ADLumin-3, and how is its affinity to Abs?*

RE: We really appreciate your positive comments on our manuscript. We agree that the binding affinity of ADLumin-1 to A β s is not very high; however, our data suggested that the affinity is suitable for both in vitro and in vivo imaging. We are still working on optimization of ADLumin-1 with the hope of stronger binding. We have tested the binding affinity of ADLumin-3 to A β s. and the K_d was $2.9 \mu M$, which is slightly weaker than ADLumin-1. We have added this information into SI Fig. 2D.

2) *In figure 3B, why the chemiluminescence spectra of ADLumin-1 alone is very low? After reacting with DMSO, there should be some light released, and in addition, the chemiluminescence was dramatically turned-on with A β 40 aggregates, that's hard to understand, without the excitation light source, the energy is conserved, and the signal intensity should only depend on the concentration of ADLumin-1 and oxygen.*

RE: It is true that some light (chemiluminescence) is released in the presence of DMSO; however, as you pointed out, the intensity is low. As you know, the restricted conformation of a probe and the hydrophobicity of the residing environment of the probe have strong influence on the “turn on” effect of fluorescence²⁻⁶, and this principle could be applied to chemiluminescence. In the pure PBS solutions (10% DMSO), the carbon-carbon bond can rotate freely, which can strongly diminish the efficiency of the emission of chemiluminescence. This is why the chemiluminescence spectra of ADLumin-1 alone is very low. By contrast, upon binding to A β aggregates, the rotation is strongly restricted to provide rigid conformations, which could significantly enhance the chemiluminescence signal. In addition, our data suggests that ADLumin-1 is binding to a hydrophobic environment; consequently the chemiluminescence was notably enhanced. The significant blue-shift of emission of ADLumin-1 with A β s in the fluorescence study (SI Fig. 2A) and molecular docking results provided strong evidence for the hydrophobic binding. Our molecular docking revealed that ADLumin-1 preferred to bind to the hydrophobic groove formed by Phe19, Ala21, Val24, Asn27, and Ile31 (Fig. 3C). By contrast, the PBS solutions can't provide a hydrophobic environment for ADLumin-1 binding.

In brief, the generation of chemiluminescence is due to auto-oxidation in the presence of O₂, and the enhanced (turn-on) chemiluminescence is due to the fact that A β fibrils can restrict the rotation of the bonds in ADLumin-1 and provide a hydrophobic environment for ADLumin-1 binding. We have added related discussions in the revised manuscript (Page 18 of our manuscript).

3) *Compared with figure 3D and F, what is the major difference of oxygen or DMSO between PBS and mouse brain homogenate?*

RE: We thank the reviewer for raising this point. We did the experiment under the same oxygen level and DMSO (10%) in both PBS and mouse brain homogenate groups. The higher chemiluminescence intensity in mouse brain homogenate than that of PBS solution is very likely due to the fact that brain homogenate contains more hydrophobic species. This is why the chemiluminescence difference between the absence and presence of A β aggregates in brain homogenate is smaller than that from the testing in PBS solutions.

4) *In vivo* chemiluminescence imaging with ADLumin-1 in figure 6A and figure 9A, in addition to the signal from the nose, the palm signal of 5XFAD mice seems to have a higher intensity than the WT group, quantitative data of these areas should be carefully analyzed to exclude the influence of administration dose.

RE: We thank the reviewer for raising the point. In our *in vivo* studies, the doses injected were adjusted with animal weight, and all of our injections were performed smoothly. From our experience, the higher paw/palm signal is always random. We also performed quantification for the imaging signals from palms, and found no significant differences between WT and 5xFAD mice (SI Fig.6D).

5) *In figure 9A, the eye signal of WT mice seems to have a higher intensity than 5XFAD, which is contrary to the result in figure 6A.*

RE: From our previous studies, we found that the positioning of the eye of a mouse had significant influence on imaging signals⁷. From our experience, Fig. 6C is the most reliable position, at which the whole eye is directly exposed to the camera. Operating with this proper positioning, we have achieved decent quantification data in Fig. 6D and SI Fig. 7. However, if the mouse is placed in the prone position, the signals from eyes are not reliable, because the eyes are covered (at least partially) by eye-lids.

References

1. Zhang, X., *et al.* Near-infrared fluorescence molecular imaging of amyloid beta species and monitoring therapy in animal models of Alzheimer's disease. *Proceedings of the National Academy of Sciences of the United States of America* **112**, 9734-9739 (2015).
2. Zhou, K., *et al.* Environment-Sensitive Near-Infrared Probe for Fluorescent Discrimination of Abeta and Tau Fibrils in AD Brain. *Journal of medicinal chemistry* **62**, 6694-6704 (2019).
3. Walker, A.S., Rablen, P.R. & Schepartz, A. Rotamer-Restricted Fluorogenicity of the Bis-Arsenical ReAsH. *J Am Chem Soc* **138**, 7143-7150 (2016).
4. Grabowski, Z.R., Rotkiewicz, K. & Rettig, W. Structural changes accompanying intramolecular electron transfer: focus on twisted intramolecular charge-transfer states and structures. *Chem Rev* **103**, 3899-4032 (2003).
5. Zhang, X. & Ran, C. Dual Functional Small Molecule Probes as Fluorophore and Ligand for Misfolding Proteins. *Curr Org Chem* **17**, 580-593 (2013).
6. Shin, J., *et al.* Harnessing Intramolecular Rotation To Enhance Two-photon Imaging of Abeta Plaques through Minimizing Background Fluorescence. *Angew Chem Int Ed Engl* **58**, 5648-5652 (2019).
7. Yang, J., Yang, J., Li, Y., Xu, Y. & Ran, C. Near-infrared Fluorescence Ocular Imaging (NIRFOI) of Alzheimer's Disease. *Mol Imaging Biol* **21**, 35-43 (2019).

REVIEWERS' COMMENTS:

Reviewer #1 (Remarks to the Author):

The authors have carefully reviewed the paper according to the referees' comments, thus I think it can be considered for publication.

Martina Zangheri

Reviewer #2 (Remarks to the Author):

I thank the authors for providing a revised manuscript that mostly satisfies my original comments and questions. The one issue that remains, however, is the inconsistency in the cartoons shown in Figure 7A and the TOC graphic. The authors propose based on calculations that ADLumin-1 binds across the different Abeta strands rather than intercalating in between the strands. This orientation is correct in the cartoon in Figure 7A, but not in the TOC graphic. For CRANAD-3, the authors state in the response letter that this molecule intercalates between the individual Abeta strands in fibrils. This orientation for CRANAD-3 is NOT correct in Figure 7A, but is correct in the TOC graphic. I would recommend the authors correct these inconsistencies before publication.

Jerry Yang

Reviewer #3 (Remarks to the Author):

I think the authors have addressed most of my concerns in the revised manuscript. I have only one small question. The author mentioned that ADLumin-1 is luminescent in the presence of 10% DMSO DMSO or in pure DMSO. What about other solutions?

Dear Reviewers,

Thank you for reviewing our manuscript titled “**Smart chemiluminescence probes and dual-amplification of signal for detection of amyloid beta species in vivo**” (NCOMMS-20-02566) by Yang, J. et al. Based on your thoughtful comments, we have revised the manuscript accordingly. The following is our point-by-point response to the Reviewers. The Reviewers’ comments are in italics.

Thank you for taking the time to handle our manuscript.

Sincerely yours,

Chongzhao Ran

Reviewer #2:

I thank the authors for providing a revised manuscript that mostly satisfies my original comments and questions. The one issue that remains, however, is the inconsistency in the cartoons shown in Figure 7A and the TOC graphic. The authors propose based on calculations that ADLumin-1 binds across the different Abeta strands rather than intercalating in between the strands. This orientation is correct in the cartoon in Figure 7A, but not in the TOC graphic. For CRANAD-3, the authors state in the response letter that this molecule intercalates between the individual Abeta strands in fibrils. This orientation for CRANAD-3 is NOT correct in Figure 7A, but is correct in the TOC graphic. I would recommend the authors correct these inconsistencies before publication.

RE: Thank you for pointing this inconsistency. We have changed the arrangement of the two molecules in Figure 7A accordingly. However, we didn’t provide TOC anymore, because TOC is not allowed in Nature Communications based on the editorial guideline.

Reviewer #3:

I think the authors have addressed most of my concerns in the revised manuscript. I have only one small question. The author mentioned that ADLumin-1 is luminescent in the presence of 10% DMSO DMSO or in pure DMSO. What about other solutions?

RE: We thank the reviewer for raising this point. We have preliminarily tested several very common solvents, like dichloromethane, methanol, ethanol, DMF, and DMSO. It turned out that ADLumin-1 was luminescent in DMF and DMSO but not in other solvents. It seems that ADLumin-1 is luminescent in aprotic solvents of high polarity. However, we have no clear explanation for this phenomenon. We will continue to investigate this phenomenon to find the exact mechanism in our further studies.